# Parental/Guardian–Child Physical Activity in Relation to Racial/Ethnic Inequities in the Americas: A Scoping Review

**DOI:** 10.3390/healthcare13233130

**Published:** 2025-12-01

**Authors:** Melquesedek Ferreira da Silva Almeida, João Antônio Chula de Castro, Andressa Ferreira da Silva, Diego Augusto Santos Silva

**Affiliations:** 1Research Center in Kinanthropometry and Human Performance, Department of Physical Education, Sports Center, Federal University of Santa Catarina (UFSC), Florianopolis 88040-900, SC, Brazil; melqueedfisica@gmail.com (M.F.d.S.A.); jchula@mcmaster.ca (J.A.C.d.C.); andressafs1988@gmail.com (A.F.d.S.); 2Child Health & Exercise Medicine Program (CHEMP), McMaster University, Hamilton, ON L8N 3Z5, Canada

**Keywords:** health, ethnicity, exercise, family, intergenerational physical activity, discrimination

## Abstract

**Background/Objectives**: This study mapped parent/guardian–child physical activity (PA) inequities across racial and ethnic groups in the Americas. **Method**: A systematic scoping review was conducted following the Preferred Reporting Items for Systematic Reviews and Meta-Analyses Extension for Scoping Reviews (PRISMA-ScR) and the Joanna Briggs Institute guidelines. Searches were carried out in January 2025 in the following databases: SciELO, LILACS, Web of Science, MEDLINE, EMBASE, Scopus, Web of Science, SPORTDiscus, and PsycINFO. Eligible studies were original articles that jointly assessed parents/guardians and children from diverse racial and ethnic backgrounds. **Results:** 4195 articles were found, and a total of 25 studies were included (cross-sectional design n = 20). Among these, 18 studies reported that higher parental/guardian PA was associated with greater child PA, regardless of race and ethnic background. Only seven studies provided explicit race and ethnic comparisons (six cross-sectional and one cohort). In four studies Black/African American parents/guardians and children were less physically active than their White counterparts; one study of Latino families showed lower participation in sports compared with African Americans; in one study Hispanic, African American, and Asian families had lower odds of engaging in PA compared with White peers; and one study of African American and Mexican American families reported lower participation in non-competitive individual sports than Anglo families. **Conclusions**: This review demonstrates consistent associations between parental/guardian and child PA, although a few studies reported null or inverse findings. Moreover, racial and ethnic inequalities in intergenerational PA reflect broader structural health inequities, where access to time, space, and resources for movement remains unevenly distributed.

## 1. Introduction

Regular physical activity (PA) is essential for the promotion of health for all [1], and its benefits for children and adolescents are a key element for adequate growth and development [2]. Moreover, participation in moderate- and vigorous-intensity PA can prevent gastrointestinal diseases (e.g., risk of gastroesophageal reflux, gastric ulcer, chronic gastritis, irritable bowel syndrome, cholecystitis, cholelithiasis, and acute and chronic pancreatitis) [3], depression [4], obesity [5], type II diabetes, hypertension, cancer, and Alzheimer’s disease [6]. In response to the low prevalence of compliance with the PA recommendations established by the World Health Organization (WHO), the World Health Assembly (WHA) approved in 2018 the Global Action Plan on Physical Activity 2018–2030, with the goal of increasing PA participation by 15% among children, adolescents, and adults by 2030. Studies conducted in the Americas showed lower PA levels among children and adolescents, both in South and Caribbean countries and in North America [7,8]. In the Latin American and Caribbean context, the results of the report indicate that less than 20% of adolescents meet the global recommendations of 60 min of moderate-to-vigorous PA daily, with particularly low prevalences in South American countries [7]. Similarly, data from the Canadian Health Measures Survey (CHMS) and the National Health and Nutrition Examination Survey (NHANES) indicate that, in North America, only a small proportion of children and adolescents meet the recommendations for daily PA, with significant decreases observed as age increases [8].

To achieve this, the plan recommends a set of 20 policy areas, all aimed at creating societies that are more physically active through the improvement of environments and opportunities for people of all ages, classes, races and ethnic backgrounds, and genders [9]. In this regard, monitoring current PA levels, trends, and influencing factors is essential for tracking health indicators and achieving the global target of increasing PA levels [10].

Among the current influencing factors on PA, parents/guardians are frequently considered significant social agents for the promotion of PA among children and adolescents [11]. Parental promotion of PA can take tangible or intangible forms [12]. Tangible support includes providing transportation or financial resources for participation in PA programs, as well as modeling, participating in, and supervising children’s PA. Intangible support includes encouragement and praise for engaging in PA [12]. Thus, parental/guardian PA and preferences are individual-level factors that can influence children’s PA [12].

Parental and guardian PA may be influenced by racial and ethnic inequities, which in turn can affect children’s PA. While this review is based on observational evidence and does not establish causal mechanisms, concepts from decolonial thought and biopolitical theory [13,14] provide a useful interpretive lens to understand how historical and structural processes have shaped unequal access to health-promoting resources across the Americas. From this perspective, racial and ethnic inequities in PA may reflect the long-standing effects of coloniality and social stratification, which continue to influence opportunities for PA and well-being among families. Structural racism operates across generations by shaping neighborhood conditions, economic opportunities, and access to safe recreational spaces, thereby reproducing social disadvantage and limiting both parents’ and children’s opportunities for PA [15]. In this context, race and ethnic minority populations have historically been subjected to social structures that marginalize them and expose them to precarious living conditions [16], directly affecting access to PA [17]. The influence of race and ethnic background on PA has been observed in the Americas, producing direct effects whereby self-identified White participants, compared with African American/Black participants, showed a lower likelihood of being physically active, resulting in what is known as race and ethnic inequities in health indicators [18]. This perspective is still under development in the health sciences literature [19], with the aim of understanding how race and ethnic inequities affect individual health [18]. Consequently, primary studies on parental/guardian and child PA that examined race and ethnic background have been developed to elucidate how race and ethnic factors affect PA among parents/guardians and children [20], highlighting the need for research specifically targeting race and ethnic minority populations [21].

Original studies developed in Europe have shown that race/ethnicity influences the PA of parents/caregivers and children [22,23]. A cohort study observing different ethnic groups shows different results for vigorous PA, and how inequities in PA reflect in childhood obesity rates [22]. Although previous reviews have investigated parental/guardian and child PA [11,24], their findings indicated that (i) parental support for children’s PA participation exerted a positive social influence, such that children and adolescents achieved higher PA levels [11]; and (ii) age, race and ethnic background, and self-concept were identified as the main factors influencing parental/guardian and child PA [24]. However, these studies did not address race and ethnic inequities in PA [11,24]. To date, no systematic or scoping reviews, published or ongoing, have analyzed race and ethnic background and inequities in parental/guardian and child PA.

Accordingly, the objective of this scoping review was to map studies conducted in the Americas that investigated the association between parental/guardian PA (fathers, mothers, or other caregivers) and child/adolescent PA across different race and ethnic groups, with the aim of analyzing the existing race and ethnic inequities.

## 2. Materials and Methods

This scoping review followed the checklist of the Preferred Reporting Items for Systematic Reviews and Meta-Analyses Extension for Scoping Reviews (PRISMA-ScR) [25] and the recommendations of the Joanna Briggs Institute [26]. The synthesis of the results followed the PAGER framework (Patterns, Advances, Gaps, Evidence for Practice, Research Recommendations) (Appendix A) [27]. The PCC (Participants, Concept and Context) strategy was used to establish the inclusion and exclusion criteria for the studies. The review was prospectively registered in the Open Science Framework (OSF), and the registration is available at the following link: “https://osf.io/mx5kj/overview (accessed on 29 January 2025)”.

### 2.1. Inclusion Criteria

#### 2.1.1. Participants

Studies whose sample of participants included in the same analysis father(s) and son(s) (children/adolescents), and/or mother(s) and daughter(s) (child/adolescent), and/or any person representing the caregiver of the child and/or adolescent were considered for this review. The age range used was 0 to 19 years, with 10 to 19 years being considered adolescent [28].

#### 2.1.2. Concept

For this review, studies were considered if they investigated race and ethnic background and race and ethnic inequities in the PA of parents/guardians and children in any region of the Americas (South, Central, or North America).

The race is conceptualized as a socially constructed category linked to structural power relations, whereas ethnicity refers to shared cultural, linguistic, or ancestral characteristics within a group [29]. With respect to the investigation of race and ethnic background, eligible studies were required to explicitly identify the race and ethnic groups analyzed in the methods section, rather than treating these categories solely as covariates or adjustment variables. No restrictions were applied regarding the inclusion of studies, provided they clearly examined parental/guardian and child PA across different race and ethnic groups.

Inequality is defined as an observable difference in health-related outcomes or behaviors between groups, whereas inequity refers to differences that are unjust and avoidable, reflecting systemic disadvantage [30]. Regarding race and ethnic inequities, eligible studies needed to include two or more race and ethnic groups to enable comparison of PA outcomes between parents/guardians and children [31].

#### 2.1.3. Context

PA can be classified as structured or incidental and described across four dimensions: mode or type of activity, frequency, duration, and intensity, which can be categorized into different levels [32]. Current PA recommendations emphasize that all adults should engage in 150 to 300 min of moderate-to-vigorous intensity PA per week, or 75 to 150 min of vigorous-intensity PA per week [33]. For adults, PA domains are classified as leisure, transportation, occupational, and household activities [34].

For children and adolescents, the recommendation is to perform an average of 60 min per day of moderate-to-vigorous intensity PA, primarily aerobic, throughout the week, in addition to vigorous-intensity aerobic activities and muscle- and bone-strengthening activities at least three days per week [28]. Considering the context in which it occurs, PA in childhood and adolescence can be analyzed in the domains of organized PA, non-organized PA, active transportation, and active tasks. However, as these measures often rely on self-reported data, both recall and social desirability biases may influence the accuracy of reported behaviors [35].

For this review, only studies that examined the association between parental/guardian and child PA were included.

### 2.2. Types of Sources of Evidence

This scoping review considered original studies published up to January 2025, the date on which the searches were conducted without temporal cut. Studies published in peer-reviewed scientific journals that presented primary data, including cohort studies, case-control studies, analytical cross-sectional studies, randomized trials, pilot studies, and interrupted time-series studies, were included in this review. Review articles, case reports, opinion pieces, conference abstracts or presentations, as well as theses, dissertations, and undergraduate final projects, were not included in this review.

### 2.3. Search Strategy

To identify studies already published in scientific databases, terms from the Medical Subject Headings (MeSH) and the Health Sciences Descriptors (DeCS) were consulted for the development of the search strategy, in addition to keywords selected by consensus from scientific sources on the topic.

To meet the proposed objectives, the authors collaborated in the initial development of the search strategy by discussing and selecting the most appropriate terms. Subsequently, two meetings were held with a specialized librarian. In the first meeting, attended by two authors, the descriptors and keywords to be included or excluded were discussed. The second meeting was dedicated to refining the strategy and defining the databases. The librarian then provided the final search strategy, structured in Word and Excel files, containing the specific commands for each database (Appendix B). A preliminary search was conducted in the MEDLINE database to assess the scope of the topic and validate the strategy.

Systematic searches were then carried out in the following databases: MEDLINE (via PubMed), Web of Science, Scopus (via Elsevier), SPORTDiscus (via EBSCOhost), Latin American and Caribbean Health Sciences Literature (LILACS, via Virtual Health Library), Scientific Electronic Library Online (SciELO), PsycINFO (via American Psychological Association—APA), and Excerpta Medica Database (EMBASE, via Ovid), in January 2025. The searches were primarily conducted in English, except for LILACS and SciELO, which were searched in Portuguese, English, and Spanish. However, no studies were found in languages other than English.

### 2.4. Study Selection/Sources of Evidence

The records identified in the databases were imported into the systematic review software Covidence^®^ (Veritas Health Innovation, Melbourne, Australia, https://www.covidence.org). Duplicates were removed partly by the software and partly through manual verification. No temporal restriction was applied for study inclusion.

The selection process was carried out in three stages. First, titles and abstracts were screened according to predefined inclusion and exclusion criteria. At this stage, the reviewers met weekly to discuss and resolve conflicts identified through the software. Second, potentially eligible studies were assessed in full text. Third, data extraction was initiated for the included studies. After completing all steps, the reference records were organized using Zotero^®^ reference manager, version 5.0.96.2 (Roy Rosenzweig Center for History and New Media, Fairfax, VA, USA).

Title/abstract screening, eligibility assessment, and examination of reference lists were conducted independently by pairs of reviewers. Discrepancies between reviewers at each stage of study selection were resolved through consensus meetings, and when necessary, the opinion of a third reviewer was sought.

### 2.5. Data Extraction

The data extraction process was conducted independently by three reviewers, who collected the following information using the data extraction tool in Covidence^®^ (Veritas Health Innovation, Melbourne, Australia). The extracted information was exported to an Excel^®^ spreadsheet (Microsoft©, Redmond, WA, USA) in a standardized format specific to this study: (1) bibliographic information (year of publication and authors); (2) country of the study; (3) study design; (4) study objective; (5) participant characteristics (sample size; parental/guardian characteristics [sex, mean/median age, education, socioeconomic level] and child/adolescent characteristics [sex and age]); (6) race and ethnic background (classifications used; measurement instrument/method of assessment); (7) parental/guardian PA (measurement instruments, PA classification, type of PA); (8) child/adolescent PA (measurement instruments, PA classification, type of PA); and (9) main results (association between parental/guardian PA and child PA; race and ethnic inequities in parental/guardian and child PA).

### 2.6. Data Analysis and Presentation

The information was presented through narrative synthesis and descriptive tables, covering the following characteristics of the included studies: year of publication and country of origin; study design and objectives; participant characteristics, including sample size, parental/guardian information (sex, mean or median age, education, and socioeconomic level) and child/adolescent information (sex and age); race and ethnic classification (categories used and instruments or methods of data collection); PA measures (instruments used, domains, and types of PA); and main results related to race and ethnic background and parental/guardian and child PA.

The analysis and understanding of the processes that lead to race and ethnic inequities in health are complex, as they require an interdisciplinary perspective and dialog across different fields [31]. In studies that included samples with at least two race and ethnic groups, inequity was considered present when a historically marginalized group showed lower PA levels. The information from this review was presented by considering the total number of studies included, which were subsequently organized into two groups: (i) studies that analyzed the association between parental/guardian and child PA without investigating race and ethnic inequities; and (ii) studies that reported results on race and ethnic inequities in parental/guardian and child PA.

## 3. Results

The search across eight databases resulted in 4195 studies. After duplicate removal using the Covidence^®^ software (n = 1765) and additional manual duplicate exclusions by the authors (n = 12), a total of 2430 records proceeded to the title and abstract screening stage. Of these, 102 articles were selected for full-text assessment and evaluated according to the eligibility criteria. In the end, 25 studies were included in the review (Figure 1).

### 3.1. Study Characteristics (n = 25)

Most of the studies included in this review employed a cross-sectional design (n = 20). In addition, two cohort studies, randomized clinical trials, and one non-randomized experimental study were identified (Table 1).

All studies included in the review were conducted in the United States of America (USA), with the earliest published in 1985 [36] and the most recent in 2023 [37]. The U.S. state with the largest number of studies was Texas (n = 5) [36,38,39,40,41], followed by South Carolina (n = 4) [37,42,43,44], Minnesota (n = 3) [20,45,46], North Carolina (n = 2) [47,48], California (n = 2) [49,50], Colorado (n = 2) [48,51], Ohio (n = 2) [45,49], and Tennessee (n = 2) [52,53]. One study conducted in the USA did not report the city and/or state [54] (Table 1). Additional information regarding the studies can be found in the Appendix A.

### 3.2. Participant Characteristics

The 25 studies included in this review investigated fathers, mothers, and grandparents as caregivers of children and adolescents (Appendix A). Among these, 16 studies investigated both fathers and mothers [20,37,40,41,42,43,44,45,46,47,48,50,51,53,55,56], six studies focused exclusively on mothers [39,49,52,54,57,58], one study investigated any caregiver/guardian without specifying the role [59], and one study examined only fathers [36]. Only one study adopted a broader approach by including fathers, mothers, and grandparents [38]. 15 studies investigated dyads [20,37,39,40,41,44,47,49,50,53,54,55,57,58,59], one study investigated triads [38], and nine presented inconsistent/different samples with different numbers of parents/guardians and children [36,42,43,45,46,48,51,52,56] (Appendix A).

**Table 1 healthcare-13-03130-t001:** General information on studies investigating parental/guardian and child PA and race and ethnic background (n = 25).

Authors and Year	Location/Region	Type of Study	Race/Ethnicity of Participants	Race/Ethnicity Measure	Children’s PA Measurement	Parents’ PA Measurement	Effect Size	Type of Analysis	Significant Association
**Studies examining the association between parental/guardian and child PA without investigating race and ethnic inequities (n = 18)**									
Alhassan et al., 2018 [57]	United States, Massachusetts	Randomized controlled trial	African American or Black	Reported by parents	Accelerometer	Accelerometer	*p* = 0.01	Anova	Yes
Cason-Wilkerson et al., 2015 [51]	United States, Colorado	Cross-sectional	Hispanic	NR	Questionnaire	Questionnaire	_	_	Yes
Eisenberg et al., 2014 [46]	United States, Minnesota	Cross-sectional	White, African American, east African, Asian, Hispanic, Native American or mixed/other	Self-reported	Questionnaire	Questionnaire	*p* < 0.001	Regression	Yes
Garcia et al., 2021 [38]	United States, Texas	Cross-sectional	White, Latino/Hispanic or Black	Self-reported	Questionnaire	Questionnaire	P: *p* = 0.001; G: *p* = 0.012	Correlation	Yes
Jago et al., 2004 [39]	United States, Texas	Cross-sectional	European Americans, Hispanic, African American	Reported by parents/caregivers	Telemetry	Questionnaire	*p* > 0.05	Correlation	No
Jang et al., 2016 [52]	United States, Tennessee	Cross-sectional	Korean American	Self-reported	Questionnaire	Questionnaire	*p* > 0.05	Regression	No
Nichols-English et al., 2006 [58]	United States, Georgia	Cross-sectional	Black	NR	Questionnaire	Questionnaire	*p* < 0.05	Correlation	Inverse
Pangalangan; Puma, 2024 [48]	United States, Colorado	Cross-sectional	Hispanic and non-Hispanic	Reported by parents	Questionnaire	Questionnaire	*p* < 0.001	Regression	Yes
Polley et al., 2005 [59]	United States, Oklahoma	Cross-sectional	African American Native American	Self-reported	Questionnaire	Questionnaire	*p* = 0.301	Correlation	No
Porter, 2017 [56]	United States, Florida	Cross-sectional	African American, Filipino Americans, Hispanic Americans	Reported by parents/caregivers	Questionnaire	Questionnaire	Mother: *p* = 0.02; Father: *p* = 0.025.	Anova	Yes
Ruiz et al., 2011 [53]	United States, Tennessee	Cross-sectional	Hispanic	Reported by parents/caregivers	Accelerometer	Accelerometer	LPA: *p* < 0.0001; MPA: *p* < 0.0001	Correlation	No
Sallis et al., 1988 [50]	United States, California	Cross-sectional	Anglo and Mexican American	NR	Questionnaire	Questionnaire	MOC: *p* = 0.05; MYC: *p* < 0.05	Correlation	Yes
Salvo et al., 2019 [40]	United States, Texas	Cohort study	Hispanic and non-Hispanic	Reported by parents/caregivers	Questionnaire	Questionnaire	*p* = 1.02	Regression	No
Sweeney et al., 2023 [37]	United States, South Carolina	Non-randomized experimental study	African American	Self-reported	Accelerometer	Accelerometer	*p* < 0.05	Regression	Yes
Trost et al., 1999 [42]	United States, South Carolina	Cross-sectional	African American	Self-reported	Accelerometer	Perception of parents’ activity level	*p* < 0.05	Regression	Yes
Wen; Su, 2015 [41]	United States, Texas	Cross-sectional	Latino	NR	Questionnaire	Questionnaire	*p* < 0.01; *p* = 0.03	Regression	Null
Wilson et al., 2022 [44]	United States, South Carolina	Randomized controlled trial	African American	NR	Accelerometer	Accelerometer	*p* = 0.012	Ancova	Yes
Wirthlin et al., 2020 [20]	United States, Minnesota	Cross-sectional	Hmong, Somali, African American, Hispanic, Caucasian and Native American	NR	Accelerometer	Accelerometer	*p* = 0.04	Regression	Yes
**Studies revealing race and ethnic inequities in parental/guardian and child PA (n = 7)**									
Duncan; Strycker; Chaumeton, 2015 [55]	United States, Oregon	Cross-sectional	African American (↓), Latino (↓) and White (↑)	NR	Accelerometer	Questionnaire	*p* < 0.05	Regression	Yes
Gottlieb; Chen, 1985 [36]	United States, Texas	Cross-sectional	Black (↓), Mexican American (↓) and Anglo (↑)	NR	Questionnaire	Questionnaire	*p* < 0.05	Regression	Yes
Madsen; McCulloch; Crawford, 2009 [45]	United States, Alabama, Ohio Minnesota Louisiana Oregon	Cohort	African American (↓) and Caucasian (↑)	Reported by parents/caregivers	Questionnaire	Questionnaire	NR	Regression	Yes
McMurray et al., 2016 [47]	United States, North Carolina	Cross-sectional	African American (↓), non-African American	NR	Accelerometer	Accelerometer	*p* < 0.015	Correlation	Yes
Morrison et al., 1994 [49]	United States, California, Ohio	Cross-sectional	Black (↓) and White (↑)	NR	Questionnaire	Questionnaire	*p* = 0.007	Correlation	Yes
Tandon; Zhou; Christakis, 2012 [54]	United States	Cross-sectional	White (non-Hispanic) (↑), Black (non-Hispanic) (↓), Hispanic (↓), Asian/Pacific Islander (↓) and Other (↓)	Reported by parent	Questionnaire	Questionnaire	*p* < 0.001	Regression	Yes
Trost et al., 1997 [43]	United States, South Carolina	Cross-sectional	African American (↓) and White (↑)	NR	Questionnaire	Questionnaire	*p* = 0.029	Regression	Yes

PA: physical activity; NR: not reported; ↓: ethnic or racial minority group; ↑: reference group; LPA: light physical activity MPA: physical activity moderate; MOC: mother and older children; MYC: mother and young children; P: parent; G: grandparent; “Significant association” (Yes) refers to studies reporting at least one statistically significant association between parental/guardian PA and child PA (*p* < 0.05). Studies in which all reported associations had *p* ≥ 0.05 were classified as “No”. One study presented inconsistent findings (significant for one PA domain but non-significant for others) and was conservatively classified as “Null”. Inverse = significant negative association.

Regarding children and adolescents, 12 studies investigated children exclusively [20,38,39,40,47,48,49,52,53,54,57,59], 10 studies analyzed children and adolescents jointly [36,37,41,42,43,50,51,55,58], and three studies focused exclusively on adolescents [44,46,55]. Of the 25 studies, 20 included children and/or adolescents of both sexes [20,36,37,38,39,40,41,42,43,44,46,47,48,50,51,52,53,54,56,59], while five studies investigated only female participants [45,49,55,57,58] (Appendix A).

Eleven studies did not specify the method of collecting information on the race and ethnic background of parents/guardians and children [20,36,41,43,44,47,49,50,51,55,58]. Six studies used self-reported race and ethnic identification from the participants themselves [37,38,42,46,52,59], while eight studies reported that race and ethnic information was provided by the parents/guardians of the children/adolescents [39,40,45,48,53,54,56,57] (Table 1).

The categorization of race and ethnic identity varied across the included studies. Twelve studies investigated participants identified as Hispanic/Latino [20,38,39,40,41,46,48,51,53,54,55,56]. 13 studies also analyzed African American participants [20,37,39,42,43,44,45,46,47,55,56,57,59]. In addition, six studies reported participants categorized as White [38,42,46,49,54,55] and Black [36,38,49,54,57,58].

### 3.3. Measurement of Physical Activity and Association Between Parental/Guardian and Child Physical Activity

Parental/guardian and child PA was measured in different ways across the 25 studies included. Most of the studies (n = 16) used questionnaires, of which 12 were self-administered and 4 were interviewer-administered, to assess PA in both parents/guardians and children [36,38,40,41,43,45,46,48,49,50,51,52,54,56,58,59]. Of these, 15 studies used questionnaires for children, with eight studies having the children answer the questionnaires [36,38,43,46,50,52,58,59], four studies having the parents/guardians answer about their children’s PA [40,41,48,54], and two studies not specifying who answered the questionnaires [49,56]. Six studies employed accelerometers to investigate PA in both parents/guardians and children [20,37,44,47,53,57]. Three studies assessed parental PA using a method different from that applied to children. One of these studies measured child PA using accelerometers and parental PA with self-administered questionnaires [55]; another used telemetry for child PA and questionnaires for parental PA [39]; and a third assessed child PA with accelerometers and parental PA through children’s perceptions [42] (Table 1). Additional information on PA (domains, frequency, intensity and time) and measurement instruments (questionnaire name, cut-off points and frequency of use) can be found in the Appendix A.

Among the 25 studies, 18 reported that parents/guardians who were more physically active had children who were also more physically active [20,36,38,42,43,44,45,46,47,48,49,50,51,53,54,55,56,57]. Of these 18 studies that found a direct association between parental/guardian and child PA, 14 investigated more than one race and ethnic group [20,36,38,43,45,46,47,48,49,50,54,55,56,57], while four focused exclusively on one group, including Hispanic [51,53] and African American populations [42,44] (Table 1).

Within the total of 25 studies, one reported that parental/guardian moderate-to-vigorous PA did not model child PA [58]. One study reported a null association overall, as parental time spent in household activities was negatively associated with children’s leisure-time PA, while parental leisure-time PA was positively associated with their children’s PA. [41]. Only five studies did not find any association between parental/guardian and child PA [37,39,40,52,59] (Table 1).

### 3.4. Studies Analyzing the Association Between Parental/Guardian and Child Physical Activity Without Data on Race and Ethnic Inequities (n = 18)

Of the total number of studies included, 18 investigated parental/guardian and child PA across different race and ethnic groups without presenting comparative results between groups, which did not allow for the analysis of race and ethnic inequities [20,37,38,39,40,41,42,44,46,48,50,51,52,53,56,57,58,59]. Among these, 11 studies reported a direct association between parental/guardian and child PA, including nine cross-sectional studies [20,38,42,46,48,50,53,56], two randomized clinical trials [44,57]. However, five studies did not find a direct association between parental/guardian and child PA [37,39,40,52,59], comprising three cross-sectional [39,52,59], one cohort [40], and one non-randomized experimental study [37]. Additionally, one cross-sectional study reported an inverse association between parental/guardian and child PA [58]. One study presented a negative association between parental domestic PA and children’s leisure-time PA, but parental participation in leisure-time PA was positively associated, classified as null [41] (Table 1 and Appendix A).

In the eight cross-sectional studies that found a direct association, the race and ethnic groups investigated varied. One study showed that joint participation in moderate-to-vigorous PA between parents/guardians and children from different race and ethnic groups was associated with greater weekly PA among adolescents, including sports, walking, and cycling [46]. Another study conducted with African American, Filipino American, and Hispanic American populations found that higher maternal leisure-time PA frequency was associated with greater child participation in organized PA [56]. A study of Mexican American and Anglo-American mothers and daughters reported that higher levels of maternal PA, particularly at very vigorous intensity, correlated with adolescent daughters’ PA [50]. In addition, a study including Hmong, Somali, African American, American, Hispanic, Caucasian, and Native American families found that for every additional hour of light-intensity parental PA, children accumulated approximately 25 extra minutes of daily PA [20]. Another study among Hispanic parents and children found that parental light- and moderate-intensity PA was associated with equivalent levels in their children [53]. One study reported that greater parental PA time and frequency were associated with higher child PA levels in both Hispanic and non-Hispanic groups [48]. Another study indicated that the weekly PA frequency of parents and grandparents (White, Black, and Latino) was positively correlated with children’s PA [38], while another reported that vigorous maternal PA positively influenced the PA of African American children [42]. One study examined Hispanic mothers and daughters and encouraged walking, dance, and organized sports, which led to increases in PA time and frequency for both parents/guardians and children [51] (Table 1 and Appendix A).

In the randomized clinical trial [57], a culturally adapted dance intervention with African American families increased vigorous PA and moderate-to-vigorous PA levels among mothers and children [57]. One experimental studies were also identified,. the study implemented culturally and motivationally tailored strategies for overweight African American adolescents and found increases in vigorous PA levels among both parents/guardians and children [44] (Table 1 and Appendix A).

### 3.5. Studies Analyzing Race and Ethnic Inequities in Parental/Guardian and Child Physical Activity (n = 7)

Of the 25 studies included in this review, only seven presented comparative results across different race and ethnic groups, which allowed for the analysis of race and ethnic inequities in parental/guardian and child PA [36,43,45,47,49,54,55]. Among these seven studies, four compared PA practices between African American and White parents/guardians and their children. In two of these studies, no significant correlations were found between African American parents/guardians and children, while correlations were observed only among White pairs [47,49]. Conversely, one study showed that maternal PA at moderate and vigorous intensities had a positive effect on daughters of both races (African American and White), although the effect was less consistent among African Americans [45]. Finally, another study found that maternal perception of PA was associated with increased vigorous PA among daughters (in the organized PA domain), with this association being more pronounced among White families than among African American families [43].

One study in this review investigated different race and ethnic groups (White, Hispanic, African American, and Asian) and showed that Hispanic, African American, and Asian mothers were less likely to engage in leisure-time PA with their daughters compared to White mothers and daughters [54]. Another study examined parental participation in leisure-time sports and child/adolescent participation in organized PA, observing that African American and Mexican American families reported lower participation in non-competitive individual sports compared to Anglo American families [36]. A further study analyzed moderate and vigorous intensity PA as well as a latent sport factor (participation in teams in the previous year, self-reported sports activities, and parent-reported sports practice), reporting significant results only for the association between African American parents/guardians and their children, with no association observed among Latino families. In addition, Latina girls demonstrated lower participation in sports compared to African American girls [55].

## 4. Discussion

This scoping review synthesized evidence from 25 studies that examined the association between parental/guardian and child PA across different race and ethnic groups. The main findings were as follows: (i) in 18 of the identified studies, parental/guardian PA was associated with higher levels of PA among children, regardless of the PA domain or intensity assessed; and (ii) among the 25 studies included, only seven specifically aimed to analyze race and ethnic differences in PA between parents/guardians and children. These seven studies identified race and ethnic inequities, highlighting distinct patterns among White, African American/Black, Asian, and Latino populations.

### 4.1. Parental/Guardian and Child Physical Activity

Most of the studies reviewed (n = 18) reported that physically active parents/guardians had children who were also more physically active, regardless of racial or ethnic background. This finding has been widely documented in the literature, indicating a significant association between the PA levels of parents/guardians and their children [11,60]. Evidence suggests that this association may be explained by multiple mechanisms, including behavioral modeling, parental support, and the joint organization of active practices within family routines [60]. In addition, active parents/guardians are more likely to value and encourage PA, provide resources and opportunities for their children to be active, and integrate PA into daily family life [11,60].

While most studies reported a positive association between parental and child PA, two studies identified an inverse relationship between African American parental and child activity levels. Specifically, higher engagement of parents in domestic or household-related PA was associated with lower levels of children’s leisure-time or recreational PA [41]. Another study also found an inverse association between vigorous parental PA and children’s PA levels [58]. Together with the findings on domestic PA, this suggests that at certain intensities and within specific domains, it may not promote shared PA with their children [41,58]. Domestic activity often reflects constrained opportunities for leisure and exercise, particularly among families facing socioeconomic disadvantage or unequal distribution of household labor [61]. Thus, these results may capture the unequal burden of unpaid work [61] and limited access to recreational opportunities, which could indirectly restrict children’s participation in organized or leisure-time PA [62].

This review also demonstrated that PA in studies on the relationship between parents/guardians and children has been investigated predominantly through self-administered questionnaires (n = 16). Questionnaires are considered valid, reliable, and cost-effective measures, with good capacity to capture information on PA domains and intensities [63]. However, these instruments may be subject to recall bias and may either underestimate or overestimate PA levels [63]. This limitation could explain the lack of association between parental/guardian and child PA, since in five of the five studies that did not identify such an association, PA was assessed using questionnaires [39,40,52,59]. Conversely, only six studies in this review employed accelerometers to investigate PA in parents/guardians and children within the same study design [20,37,44,47,53,57]. Accelerometers can generate more accurate information, particularly regarding PA intensity [64]. In five of the six studies using accelerometers, a direct association was found between parental/guardian and child PA [20,37,44,47,53,57], suggesting that when PA is measured through device-based methods such as accelerometry, there is strong evidence of a direct association between the PA of parents/guardians and their children.

### 4.2. Racial and Ethnic Inequities in Parental/Guardian and Child Physical Activity

In this review, seven studies were identified that reported comparative results across different racial and ethnic groups, allowing for the examination of racial and ethnic inequities in parental/guardian and child PA. These findings showed that African American, Hispanic, Mexican American, and Asian populations presented lower levels of PA compared to majority racial and ethnic groups, particularly White populations. The reviewed studies consistently indicated that these inequities may be influenced by contextual factors such as socioeconomic disadvantage, unsafe neighborhoods, limited access to recreational spaces, and cultural norms that restrict minority participation in organized PA [65]. Such barriers may attenuate the intergenerational transmission of active behaviors from parents to children.

Although the literature consistently documents lower PA levels among racial and ethnic minority populations, highlighting inequities in PA between parents/guardians and children, discussions regarding the structural causes of these inequities remain insufficiently developed [50,54]. From a theoretical perspective, these intergenerational inequities can be interpreted through the lens of structural racism, which reproduces social disadvantage across generations, limiting both parents’ and children’s opportunities for PA [15]. This understanding aligns with frameworks of social determinants of health, emphasizing that racial and ethnic inequities in PA are not merely behavioral but rooted in historical and institutional processes [13].

In this review, the term “racial/ethnic inequities” or “racial and ethnic inequities” did not appear explicitly in any of the studies, which can be considered a limitation in the field, given that progress toward racial and ethnic equity in health cannot be achieved if such inequities are not made explicit in the texts. This omission may reflect the structural racism affecting the Americas, a region marked by a colonial legacy [31]. Within this context, inequities in PA among parents/guardians and children not only mirror broader social determinants but also perpetuate intergenerational disadvantages in access to health and well-being opportunities [66]. This scenario can be partly explained by the historical subordination imposed on racial and ethnic minority groups, who continue to face adverse conditions such as greater exposure to stress, employment in lower-status occupations, heavier workloads, limited access to public recreational spaces, racism, and the historical residential segregation among racial and ethnic groups across the Americas [65]. These mechanisms help explain why the correlations between parental and child PA were generally weaker among minority families compared to White families, as structural and environmental constraints may override behavioral modeling effects. Comparable inequities have been identified in other global contexts, for instance, in Europe [67], where racial and ethnic families similarly experience structural and cultural barriers to participation in PA. This perspective reinforces that inequities in PA are not confined to the Americas but represent a wider pattern of social exclusion that transcends regional boundaries.

The findings of this review are also promising in terms of identifying potential PA strategies with greater adherence and engagement among families from racial and ethnic minority groups. One of the studies included in this review reported higher adherence and engagement in culturally adapted dance programs targeting African American mothers and daughters [57]. This result reinforces the notion that historical and cultural appreciation may be essential components for the success of interventions promoting PA among parents/guardians and children [31]. Such culturally grounded approaches highlight that incorporating community traditions and local identities can enhance participation in, and the long-term sustainability of, PA interventions across generations. Consequently, advancing the development of more inclusive, effective, and equitable PA programs becomes feasible, thereby enhancing the positive impact of public health interventions as such programs are culturally adapted [31].

This scoping review mapped evidence on the associations between parental/guardian and child PA across different racial and ethnic groups in the Americas. The main limitations of this review include its reliance on the available literature; although only seven studies provided direct racial/ethnic comparisons, the available evidence suggests consistent patterns of lower PA levels among racial and ethnic minority populations compared with majority groups. There is a heterogeneity in the instruments used to measure PA, limiting the comparison of results. However, the limited number of comparative analyses prevents definitive conclusions about racial and ethnic inequities. Rather than confirming inequities, the findings highlight a significant research gap: the underrepresentation of racial and ethnic dimensions in studies on intergenerational PA. Although searches were conducted in languages such as Portuguese and Spanish in the databases that allowed for such searches, no articles were found other than in English. One of the studies that presented results for inequities included the reference values in the body of the text [45]. None of the studies that yielded results regarding inequalities were of the experimental type. No studies have investigated the influence of all domains of PA. This restricts the conclusions to the data identified and the absence of studies directly addressing racial and ethnic inequities in PA between parents/guardians and children. The geographical concentration of studies in North America, particularly in the United States, also limits the generalizability of the findings to other sociocultural contexts, such as South and Central America, requiring caution in the interpretation of results. Nevertheless, this review is strengthened by integrating three constructs within a single research framework: (1) intergenerational parent/guardian–child relationships; (2) PA; and (3) racial and ethnic inequities. This combination situates the study within the broader discussion on the interconnections between parent/guardian and child health and social inequities in PA.

This scoping review also identified potential avenues for future research in this field. Future research examining the associations between parental/guardian and child PA across different racial and ethnic groups in Central and South America is warranted, as no studies were identified in these regions. Furthermore, studies should not only compare PA levels across racial and ethnic groups but also explore mediating mechanisms, such as parental role modeling, neighborhood safety, discrimination experiences, and socioeconomic stress, which influence intergenerational patterns of PA. Also, studies need to clarify the influence of PA in different domains (work, household activities, and transportation) on the PA of children. Additionally, longitudinal and experimental designs are needed to examine causal pathways and the effectiveness of culturally tailored interventions in promoting equity. For instance, studies employing observational and experimental designs are needed to investigate racial and ethnic inequities across different regions of the Americas, with particular attention to the legacies of historical contexts. In sum, this review underscores that addressing racial and ethnic inequities in parental and child PA requires integrating social, cultural, and environmental perspectives into research and policy, thereby advancing the goal of intergenerational health equity.

## 5. Conclusions

The majority of studies in this review demonstrated positive links between parental/guardian and child PA, while a few studies reported null findings, reflecting domain-specific variations in how parental influence manifests, and one identified an inverse association specifically for vigorous-intensity PA. The findings indicate that parental PA positively influences children’s PA, regardless of race or ethnicity, reinforcing the importance of family dynamics in promoting active behaviors. However, racial and ethnic inequalities in intergenerational PA reflect the materialization of structural health inequalities, in which access to time, space, and resources for movement are unequally distributed. Historically minoritized groups, such as Black/African American, Hispanic, Mexican American, and Asian American individuals, presented lower levels of PA compared to White individuals. Furthermore, within these racialized groups, Latinos tended to have even more unfavorable outcomes than African Americans, which may reflect additional layers of inequality and structural repression among ethnic and racial subgroups.

## Figures and Tables

**Figure 1 healthcare-13-03130-f001:**
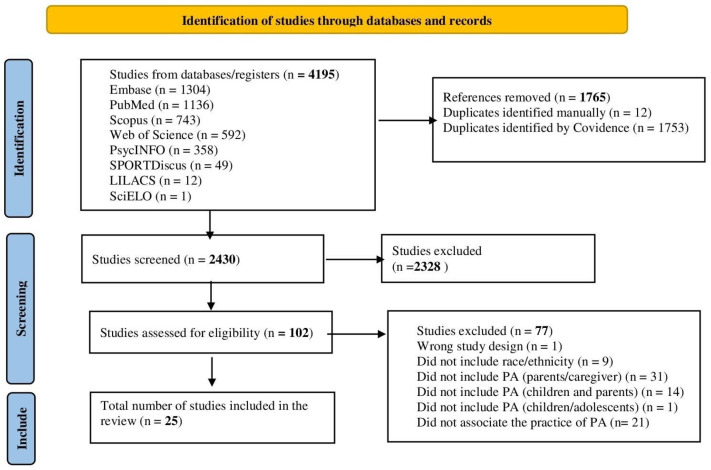
Flowchart PRISMA-ScR of the scoping review of parental/guardian–child PA in relation to racial/ethnic inequities in the Americas.

## Data Availability

No new data were created or analyzed in this study.

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
