# Peer review of "Parental/Guardian–Child Physical Activity in Relation to Racial/Ethnic Inequities in the Americas: A Scoping Review"

_healthcare, 2025, doi:10.3390/healthcare13233130_

Round 1

Reviewer 1 Report

Comments and Suggestions for Authors

General comments: 

The title has to be reorganized for clarity because it contains too many "ands". 

There's no need to start with highlights (journal guideline). 

Abstract: Include databases and a period of time encompassed by the search (starting year chose to be included in this review). Connect the findings to practical implications and the potential impact on public health. The findings should be transparent on the study design, whether it was cross-sectional or longitudinal. 

Introduction: There is a lack of evidence from the Americas supporting the prevalence of PA for children and adolescents. 

Introduction: There is need to be more references supporting the aim of the study, not only from the Americas, but also from other developed/developing countries. The authors also provided references, with none of these are addressed thoroughly. Are they cross-sectional or longitudinal? 

Method: The inclusion criteria should include the age of children and the time frame that the search covered (the starting year was selected to be included in this review). 

Results: Every study, including experimental and RCT studies, needs to have a thorough description. 

The discussion is brief and of poor quality as it does not explain the research findings and presents several possible explanations for the associations. 

To improve readability, arrange the limitations in an understandable and methodical way. Give more precise information about the limitations, including the types of challenges encountered and their effects on the research. Provide additional precise information about the challenges encountered, such as the differences in data collection techniques and any potential biases. 

Give specific recommendations for further study and interventions. Make the information more structured by keeping the recommendations and findings apart. Consider carefully the findings' ramifications and how they might affect subsequent interventions. 

Kindly adhere to the journal guideline for references.

Comments on the Quality of English Language

The manuscript requires significant editing by a native English speaker.

Author Response

Evaluator 1

General comments:

Comment 1: The title needs to be reorganized for clarity, as it contains too many "and"s.

Response 1: We appreciate the suggestion to change the article title; however, we have chosen to keep it as is: “Physical activity of parents/guardians and children in relation to racial/ethnic inequalities in the Americas: a scoping review”.

Comment 2: There is no need to start with the highlights (magazine guideline).

Response 2: Thank you for your feedback. As the newspaper does not require this information, we have chosen to remove the topic.

Comments 3: Summary: Include the databases and time period covered by the research (the initial year chosen for inclusion in this review). Connect the findings to practical implications and potential public health impact. Results should be transparent regarding the study design, whether cross-sectional or longitudinal.

Response 3: Thank you for the suggestion. We have already added the information to make the summary clearer.

-The databases in which the research was conducted have been added, as well as the dates of the research (Lines: 15 and 16).

-We outlined the study's design in the abstract (Lines: 21-22).

-The results were linked to health implications (Lines: 28-32).

Comment 4: Introduction: There is a lack of evidence in the Americas to support the prevalence of physical activity in children and adolescents.

Response 4: Thank you for your observation. We have added the prevalence among the regions of the Americas to the first paragraph of the introduction (Lines: 45-53).

“Studies conducted in the Americas show low levels of physical activity among children and adolescents, both in South American and Caribbean countries and in North America. In the Latin American and Caribbean context, the report’s results indicate that less than 20% of adolescents meet the global recommendations of 60 minutes of moderate to vigorous physical activity per day, with particularly low prevalences in South American countries [7]. Similarly, data from the Canadian Health Measures Survey (CHMS) and the National Health and Nutrition Examination Survey (NHANES) indicate that, in North America, only a small proportion of children and adolescents meet the recommendations for daily physical activity, with significant decreases observed with increasing age.”

Comment 5: Introduction: More references are needed to support the study's objective, not only from the Americas but also from other developed/developing countries. The authors also provided references, but none of them are fully addressed. Are these cross-sectional or longitudinal studies?

Response 5: We appreciate the reviewer's suggestion. Two more studies, both from Europe, with different designs, have been included: a cohort study and a cross-sectional study (Lines: 81-84).

Original studies conducted in Europe have demonstrated that race/ethnicity influences the physical activity of parents/caregivers and children. A cohort study observing different ethnic groups shows distinct results for vigorous physical activity and how inequalities in physical activity are reflected in childhood obesity rates.

Comment 6: Method: Inclusion criteria should include the children's age and the period covered by the research (the initial year was selected for inclusion in this review).

Response 6: We appreciate the reviewer's suggestion. The information can be found in section 2.1.1.

Comment 7: Results: All studies, including experimental studies and randomized clinical trials, need a complete description.

Response 7: We appreciate the suggestion and understand the need to present the studies included in the review, all of which are rich in information. Therefore, we have created Supplementary Table 3, which presents information on the objective, methodological details, and main results of all studies, including experimental ones, as suggested.

Comment 8: The discussion is brief and of low quality, as it does not explain the research results and presents several possible explanations for the associations.

Response 8: Thank you for your comment. We have made significant modifications to all paragraphs of the Discussion to further develop it and highlight the results obtained in the review (P 14-15; Lines: 412-438, 445-450, 454-459, 452-458, 472-479, 486-488, 492-508, 519-528).

Comment 9: To improve readability, organize the limitations in a clear and methodical way. Provide more precise information about the limitations, including the types of challenges encountered and their effects on the research. Present additional precise information about the challenges encountered, such as differences in data collection techniques and any potential biases.

Response 9: We appreciate the valuable suggestion; therefore, we have made some additions to address the identified gaps (Lines: 490-505). The PAGER model (Patterns, Advances, Gaps, Evidence for Practice, Research Recommendations) has also been added for scoping reviews in Supplementary Table 4.

Comment 10: Provide specific recommendations for future studies and interventions. Structure the information better by separating recommendations from conclusions. Carefully consider the implications of the conclusions and how they may affect subsequent interventions.

Response 10: Thank you for your comment. We have made some changes and also separated the paragraphs relating to the limitations (P15; Lines: 519-528).

This exploratory review also identified potential avenues for future research in this area. It is necessary to investigate the associations between the physical activity of parents/caregivers and children in different racial and ethnic groups in Central and South America, as no studies were identified in these regions. Such research should not only compare physical activity levels across racial and ethnic groups but also explore mediating mechanisms, such as parental example, neighborhood safety, experiences of discrimination, and socioeconomic stress, that influence intergenerational patterns of physical activity. It is also necessary to clarify the influence of physical activity in different domains (work, domestic activities, and transportation) on children's physical activity. Furthermore, longitudinal and experimental studies are needed to examine causal relationships and the effectiveness of culturally adapted interventions in promoting equity.

Comment 11: Please follow the journal's guidelines for citations.

Response 11: Thank you for your feedback. We have made the necessary corrections.

Reviewer 2 Report

Comments and Suggestions for Authors
  1. The title and abstract claim coverage of the “Americas,” but the included studies are exclusively U.S.-based. I recommend aligning the scope stated in the title/abstract/discussion with the empirical scope (United States) or, at minimum, explicitly noting the reasons for the failed retrieval of Latin American studies and plans for additional searches. If you retain the broader scope, the abstract and limitations must clearly state that only U.S. studies were identified within the Americas.

  1. Greater precision is needed in defining and operationalizing “inequity.”

(2-1) Only seven studies provided direct racial/ethnic comparisons; the remaining nineteen offered no comparative statistics, making any determination of inequity impossible. Soften the claims in the conclusions and highlights from “inequity is confirmed” to “evidence is limited due to the small number of comparative studies,” and add language indicating the need for further research and/or reanalysis of primary datasets.

(2-2) The manuscript does not clearly distinguish inequity from inequality, specify statistical thresholds/effect-size cutoffs, or define the reference group. Define inequality as an observed difference and inequity as an unjust/remediable difference, and state that this scoping review is limited to observing inequalities. Provide explicit decision rules (e.g., effect size (|r|), OR, β with 95% CI and (p<.05) as minimal criteria). Fix the reference group (e.g., the majority/White group) to standardize comparisons, and use unified arrows (↓/↑) to indicate directionality in tables. Supply an appendix table detailing racial/ethnic categorization, metric harmonization (minutes/day, per week, MVPA, accelerometer cut-points), and statistical standardization rules.

  1. Use of theory should be restrained. Clarify how macro-level frameworks (necropolitics/biopower) concretely inform interpretation of the results. Because the evidence base is largely observational and correlational, macro-causal claims are not warranted. Proximate explanations—SES, neighborhood safety/facility access, time poverty, cultural norms—may be more direct. The current text does not operationalize the decolonial lens via measurable variables (e.g., discrimination indices, public-investment gaps, park access). If these links cannot be articulated, down-scope this section and frame it cautiously as a possibility rather than a conclusion, avoiding over-generalization for a scoping review.

Example: “Because this review is based on observational studies, it does not establish causality. However, decolonial/biopower perspectives may hypothesize that historical patterns of policy and infrastructure distribution contribute to contemporary resource inequalities.”

  1. Address measurement heterogeneity. Results are split by device-based (accelerometry) versus self-report; the overall conclusions should likewise differentiate the strength of evidence by measurement type.

  1. Search and protocol transparency. Present databases, time frames, languages, and full search strings in a table to facilitate replication. Provide preregistration details (e.g., OSF) in the main text and appendix, including registration timing and versioned links.

  1. Rationale for the absence of non-English studies. Given searches in LILACS/SciELO, explain concretely why non-English studies were not retrieved (e.g., keyword choices, indexing lags). If feasible, add a sensitivity check removing language filters and report the results.

  1. PRISMA flow consistency. Verify that stage-wise counts in text and figure match; some numbers appear inconsistent. Add a bar chart summarizing the top exclusion categories.

  1. Discussion. Distinguish structural factors (safety, facility access, time poverty) from within-household factors (parental modeling, joint participation), and organize the argument accordingly.

  1. Conclusions. Parent–child PA associations are generally consistent, but claims regarding racial/ethnic inequities are weakly supported. State the limitations clearly (small number of comparative studies, heterogeneous measures), keep policy implications tentative, and recommend follow-up quantitative syntheses or meta-analyses.

  1. Minor. Standardize terminology: use a single, consistent scheme for inequity vs. inequality and race vs. ethnicity throughout.

Author Response

Comments 1: The title and abstract claim coverage of the “Americas,” but the included studies are exclusively US-based. I recommend aligning the scope stated in the title/abstract/discussion with the empirical scope (United States) or, at minimum, explicitly noting the reasons for the failed retrieval of Latin American studies and plans for additional searches. If you retain the broader scope, the abstract and limitations must clearly state that only US studies were identified within the Americas.

Response 1: We thank the reviewer for the observation. However, we respectfully consider that the title and scope remain consistent with the methodological framework of this scoping review. As described in the Methods section, the inclusion criteria were designed to encompass studies conducted across all regions of the Americas. The fact that eligible studies were conducted in the United States reflects an evidence gap rather than a methodological limitation.

This gap was explicitly acknowledged in the Results and discussed in more detail in the Discussion section, where we emphasized the need for future research in Central and South America to address racial and ethnic inequalities in physical activity among parents and children. Maintaining the broader geographic reference in the title (“in the Americas”) is therefore essential to highlight the review’s intention to map the evidence across the region and expose where such evidence is currently scarce—which is precisely one of the main contributions of this study.

Comment 2: Greater precision is needed in the definition and operationalization of "inequality".

(2-1) Only seven studies provided direct racial/ethnic comparisons; the remaining nineteen offered no comparative statistics, making any determination of inequality impossible. Soften the statements in the conclusions and highlights, from “inequality is confirmed” to “evidence is limited due to the small number of comparative studies,” and add language that indicates the need for more research and/or reanalysis of the primary datasets.

Response 2: We thank the reviewer for this important observation. The goal of a scoping review is to identify, map, and summarize the extent and nature of the evidence on a given topic, including existing knowledge gaps. In this context, the limited number of studies that stratified analyses by race/ethnicity reinforces the review's main conclusion: that the field has not systematically investigated racial and ethnic inequalities in intergenerational physical activity. Thus, the apparent scarcity of comparative evidence strengthens, rather than weakens, the rationale and contribution of this review. As correctly observed, only seven of the included studies provided direct racial/ethnic comparisons, while the others mentioned race/ethnicity without stratified analyses. Therefore, the current evidence base does not allow for definitive conclusions about inequalities. Through this observation, we have made the following addition to the discussion (P15; Lines: 495-508):

This exploratory review mapped the evidence on the associations between physical activity (PA) of parents/caregivers and children in different racial and ethnic groups in the Americas. The main limitations of this review include its reliance on the available literature; although only seven studies provided direct racial/ethnic comparisons, the available evidence suggests consistent patterns of lower PA levels among racial and ethnic minority populations compared to majority groups. However, the limited number of comparative analyses prevents definitive conclusions about racial and ethnic inequalities. Instead of confirming inequalities, the results highlight a significant gap in research: the underrepresentation of racial and ethnic dimensions in studies on intergenerational PA. This restricts conclusions to the identified data and the absence of studies that directly address racial and ethnic inequalities in PA between parents/caregivers and children. The geographic concentration of studies in North America, particularly the United States, also limits the generalizability of the results to other sociocultural contexts, such as South and Central America, requiring caution in the interpretation of the findings.

Conclusions (P16; Lines: 535-545)

This review comprehensively mapped the available evidence on physical activity (PA) among parents or caregivers and their children across different racial and ethnic groups in the Americas. The results indicate that parental PA positively influences children's PA, regardless of race or ethnicity, reinforcing the importance of family dynamics in promoting active behaviors. However, racial and ethnic inequalities were observed in the PA practices of parents/caregivers and their children. Historically marginalized groups, such as Black/African Americans, Hispanics, Mexican Americans, and Asian Americans, showed lower levels of PA compared to white individuals. Furthermore, within these racialized groups, Latino individuals tended to show even more unfavorable outcomes than African American individuals, which may reflect additional layers of inequality and structural repression between ethnic and racial subgroups.

Comment 3: In the revised version, we softened the language from "inequality is confirmed" to "the evidence is limited due to the small number of comparative studies." We also broadened the discussion to emphasize that this lack of comparative data itself represents a critical scientific gap in the field.

(2-2) The manuscript does not clearly distinguish between inequity and inequality, does not specify statistical thresholds/effect size cutoff points, nor does it define the reference group. Define inequality as an observed difference and inequity as an unfair/remediable difference, and state that this scoping review is limited to the observation of inequalities. Provide explicit decision rules (e.g., effect size (|r|), OR, β with 95% CI and (p<0.05) as minimum criteria). Fix the reference group (e.g., the majority/white group) to standardize comparisons and use unified arrows (↓/↑) to indicate direction in tables. Provide an appendix table detailing racial/ethnic categorization, metric harmonization (minutes/day, per week, MVPA, accelerometer cutoff points), and statistical standardization rules.

Response 3: Thank you for your suggestion. We have done as you suggested and distributed some of the requested information. Information on the distinction between difference and inequality has been added to the methods section (Lines: 133-136). The strategies used for inclusion and analysis of the studies have also been added (P5, Lines: 127-135).

In the main table (Table 1), for studies that performed comparisons between groups, we indicate the reference and/or marginalized group, and add the p-value for each study, as well as the types of analyses used.

Information regarding physical activity (frequency, intensity, duration, domain) and details about the instruments used (questionnaire name, cut-off points, questions used, brand of device used, number of days of use) are included in Supplementary Table 2. For more specific and detailed aspects of each study, Supplementary Table 3 was created, which contains information on: Objective, methods, and main results.

Comment 4: The use of theory should be moderate. Clarify how macro-level structures (necropolitics/biopower) concretely inform the interpretation of the results. As the evidence base is predominantly observational and correlational, claims of macro-causality are not justified. Immediate explanations—socioeconomic level, neighborhood safety/access to facilities, lack of time, cultural norms—may be more straightforward. The current text does not operationalize the decolonial perspective through measurable variables (e.g., discrimination indices, public investment gaps, access to parks). If these connections cannot be articulated, restrict the scope of this section and present it cautiously as a possibility, not a conclusion, avoiding overgeneralizations for a scoping review.

Example: “Since this review is based on observational studies, it does not establish causality. However, decolonial/biopower perspectives may hypothesize that historical patterns of policy and infrastructure distribution contribute to contemporary resource inequalities.”

Response 4: We appreciate the suggestion. We have revised this paragraph to address the reviewer's concern about the use of macro-level theoretical frameworks. The revised version clarifies that decolonial and biopolitical perspectives are used as interpretive lenses, not as causal explanations. The text now explicitly acknowledges that the review is based on observational evidence and that these theories serve to contextualize, not determine, observed inequalities in physical activity among racial and ethnic groups (P1, Lines: 67-78).

“The physical activity (PA) of parents and caregivers can be influenced by racial and ethnic inequalities, which, in turn, can affect children’s PA. Although this review is based on observational evidence and does not establish causal mechanisms, concepts from decolonial thought and biopolitical theory [13,14] provide a useful interpretive lens for understanding how historical and structural processes have shaped unequal access to health-promoting resources in the Americas. From this perspective, racial and ethnic inequalities in PA may reflect the long-standing effects of coloniality and social stratification, which continue to influence PA and well-being opportunities among families. Structural racism operates across generations, shaping neighborhood conditions, economic opportunities, and access to safe recreational spaces, thus reproducing social disadvantage and limiting physical activity opportunities for both parents and children.”

Comment 5: Address the heterogeneity of measurements. The results are divided between device-based data (accelerometry) and self-reported data; the overall conclusions should similarly differentiate the strength of evidence by measurement type.

Response 5: Thank you for the suggestion. We did not use the same strategy for the review discussion, as we sought to answer the study's objectives.

- To map studies conducted in the Americas that investigated the association between the physical activity of parents/guardians (fathers, mothers, or other caregivers) and the physical activity of children/adolescents in different racial and ethnic groups. What is this relationship like?

- To verify the existence of racial and ethnic inequalities in physical activity practices between parents and children in the Americas.

For this reason, the discussion only covered two topics and, also to allow for a more fluid discussion, but within the topic that deals only with the practice of physical activity without addressing inequalities, we reformulated and added more information about the measurement instruments, relating them to the results of the studies (P12-13; Line: 424-438).

Comment 6: Transparency of search and protocol. Present the databases, time periods, languages, and full search strings in a table to facilitate replication. Provide pre-registration details (e.g., OSF) in the main text and appendix, including the registration timeline and links to versions.

Response 6: Thank you for your comment. We have added the information to the main text and the appendix. The link is private to prevent access to the material before publication.

Comment 7: Justification for the absence of studies in other languages. Considering the searches in LILACS/SciELO, explain concretely why studies in other languages ​​were not retrieved (e.g., keyword selection, indexing delays). If possible, add a sensitivity check by removing language filters and present the results.

Response 7: Thank you for your comment; however, searches were conducted in all three languages ​​in the Scielo and LILACS databases, but no studies were found in any language other than English. This information has been added as one of the limitations (P15; Lines: 502-503).

Comment 8: Consistency of the PRISMA flow. Verify that the step counts in the text and figure match; some numbers appear inconsistent. Add a bar chart summarizing the main exclusion categories.

Response 8: Thank you for your feedback; the inconsistencies in the Prisma flowchart have been corrected. Additionally, a bar chart has been created and added as Supplementary Figure 1.

Comment 9: Discussion. Distinguish between structural factors (security, access to facilities, lack of time) and intra-family factors (parental modeling, joint participation) and organize the argument accordingly.

Response 9: We appreciate the reviewer's thoughtful suggestion to distinguish between structural and intrafamilial factors influencing the physical activity of parents and children. However, we respectfully consider that such a reorganization would not be consistent with the primary objective and scope of this scoping review.

The intergenerational association between physical activity of parents and children is already well established in the literature. Therefore, the focus of this review was not to revisit these behavioral mechanisms, but rather to map and critically discuss the racial and ethnic inequalities that shape these associations in the Americas.

In this sense, the discussion was strengthened by the addition of explanatory paragraphs that address structural determinants, historical contexts, and cultural elements that influence these inequalities, without restructuring the argument around intrafamilial behavioral factors (Lines: 412-423, 445-451, 454-459, 472-479).

Comments 10: Conclusions. The associations between physical activity of parents and children are generally consistent, but claims about racial/ethnic inequalities have little support. It is important to clarify the limitations (small number of comparative studies, heterogeneous measures), keep the policy implications provisional, and recommend subsequent quantitative syntheses or meta-analyses.

Response 10: We appreciate your comments and suggestions. Given the heterogeneity of the study designs and measures used, this scoping review does not allow for a quantitative synthesis. Therefore, policy implications should remain provisional, and future meta-analyses are recommended once sufficient comparable data are available. We believe these implications are of utmost importance and have included them in the text (lines 492-508, 535-545).

Comment 11: Minor. Standardize terminology: use a single, consistent scheme for inequity versus inequality and race versus ethnicity throughout the text.

Response 11: We appreciate your comment, which is extremely relevant. We have added the information to the methodology section (Lines: 125-127, 133-135).

Reviewer 3 Report

Comments and Suggestions for Authors

The authors conducted a literature review classified as a scoping review. This exempts them from synthesising the results but obliges them to map the literature thoroughly. The more detailed objectives of the review/research questions should be more clearly formulated here. The quality of scoping review could be improved by PAGER framework.

The methodology is appropriate, with a precise description of the selection strategy and the use of the PRISMA-ScR framework. The description is very detailed, even too detailed, in my opinion. The main message of this paper is lost in the flood of technical details and summaries. The results look like a balance sheet of 26 selected works according to various, but very basic criteria. There are many inaccuracies and errors in the study that undermine its trustworthiness. I present them below as major concerns.

Firstly, I do not understand why, when defining eligibility, no condition was set that the papers should concern ethnic inequalities in the PA of children and parents. There are only seven such papers.  

Secondly, the classification of studies according to study design given in the text is correct (paragraph 3.1). Table 1 shows other categories of studies that contradict the classifications of epidemiological studies. Most of the selected studies should be in the analytical category, which is divided into cohort, case-control, and cross-sectional studies.  Occasionally, non-observational experimental studies were with or without randomisation were found. I do not know why many cross-sectional studies were classified as descriptive (prevalence and incidence). It was thus assumed that these are not analytical studies and therefore cannot show the relationship between PA in children and parents.

Table 1 states that the child-parent dyad was studied. This is not apparent from the content of many articles.

It is difficult for the reviewer to find and read these 26 papers, but I see errors that undermine the credibility of this review. For example, paper [43 - Cason-Wilkerson] concerns qualitative research (focus groups) and is described as an analytical, non-randomised experimental study.

Another serious reservation is the lack of mapping of PA assessment methods. The statement that the questionnaire was used for the study is too general. The reader might have expected a precise description of the tools used for parents (even the name of the tool) and, in relation to children, a description of the questions and scales used for different age groups. Sometimes, especially regarding young children, outdoor activities were examined. I suppose there is a gap in knowledge that such a review could reveal, and that is the lack of precise measurement of activity, with a tendency to limit oneself to ad hoc questionnaires and very general questions.

And the next serious objection is the lack of a description of the statistical methods used to analyse the relationship between children's PA and that of their parents. 

Finally, the main claim (lines 19-23) does not follow from the review.

Other minor comments and editing issues include:

  • Inappropriate keywords that do not match the content. Why Racism and Necropolitics without Physical activity
  • One paragraph in Portuguese (lines 110-113).
  • Errors in the bibliography, despite the use of Zotero, e.g., in [37] two first authors combined.
  • In overall, the references list does not meet Healthcare standards.
  • Occasional typos, including in the tables in the appendix.

Author Response

Comment 1: The authors conducted a literature review classified as a scoping review. This exempts them from summarizing the results, but requires them to map the literature comprehensively. The more detailed objectives of the review/research questions should be formulated more clearly. The quality of the scoping review could be improved by using the PAGER methodology.

Response 1: We appreciate the suggestion and agree that the greater the methodological rigor, the greater the credibility and impact of a manuscript. Therefore, we have included the information related to PAGER as Supplementary Table 4.

Comment 2: First of all, I don't understand why, when defining the eligibility criteria, no condition was established that the articles address ethnic inequalities in the area of ​​children's and parents' health. There are only seven such items.

Response 2: Thank you for your comment. However, in response to your question, when establishing inclusion and exclusion criteria, studies should investigate physical activity practices by parents and children and establish an association between them, in addition to presenting race and/or ethnicity as a primary outcome. We used this criterion and included studies, for example, with only one racial/ethnic group, because we believe that compiling this data on physical activity practices by parents and children from different racial/ethnic populations in the Americas would be very important for the field of study. Our second objective was to investigate racial inequalities in physical activity practices by parents and children. To verify inequalities, the study needed to make comparisons between the groups.

Comment 3: Secondly, the classification of studies according to study design presented in the text is correct (paragraph 3.1). Table 1 shows other categories of studies that contradict the classifications of epidemiological studies. Most of the selected studies should be in the analytical category, which is divided into cohort, case-control, and cross-sectional studies. Occasionally, non-observational experimental studies with or without randomization were found. I don't know why many cross-sectional studies were classified as descriptive (prevalence and incidence). It was therefore assumed that these are not analytical studies and, consequently, cannot demonstrate the relationship between physical activity in children and parents.

Response 3: Thank you for your comment. We reviewed all the studies again. Therefore, we chose to standardize the information as described in the text.

Comment 4: Table 1 states that the child-parent dyad was studied. This is not evident in the content of many articles.

Response 4: Thank you again for your observation. Supplementary Table 1 contains all the information from studies that reported data on the dyad; some studies investigated the triad. However, we affirm that a rigorous analysis was conducted to include only studies that investigated and established the association between the physical activity of parents/caregivers and children. The gaps in the information regarding dyads demonstrate a lack of methodological clarity in the studies. Therefore, we chose to add the following information to the results (P 10-11, Lines: 268-271) , also highlighting this gap in the discussion section of the article (P15; Lines: 507-508).

Comment 5: It is difficult for the reviewer to find and read these 26 articles, but I see errors that compromise the credibility of this review. For example, article [43 - Cason-Wilkerson] deals with qualitative research (focus groups) and is described as a non-randomized analytical experimental study.

Response 5: We appreciate your observation and careful attention to the information in the studies; this greatly contributes to this article. Therefore, we chose to re-check all included studies, changing the table names to ensure they match the descriptions presented in the text. Regarding the study mentioned, there was indeed an error in the initial analysis, because, although the study is qualitative, it investigates the physical activity of parents and children in a cross-sectional, not experimental, manner. The correction has already been made.

Comment 6: Another important caveat is the lack of mapping of physical activity assessment methods. The statement that a questionnaire was used in the study is too generic. The reader might expect a precise description of the tools used for parents (including the name of the tool) and, regarding children, a description of the questions and scales used for different age groups. Sometimes, especially with regard to young children, outdoor activities were examined. I suppose there is a gap in knowledge that a review like this could reveal, and that gap is the lack of precise measurement of activity, with a tendency to limit oneself to ad hoc questionnaires and very generic questions.

Response 6: Thank you for your suggestion. Although it is not the objective of the scope review, we understand that it is important to provide this data. Therefore, information regarding all physical activity data (domain, intensities, frequency, duration, type of instrument (direct and indirect measurement), specific instrument information (equipment brand or questionnaire name), questions used, cut-off points for accelerometry use) can be found in supplementary table 2.

Comment 7: And the next serious objection is the lack of a description of the statistical methods used to analyze the relationship between the physical activity of children and that of their parents.

Response 7: Thank you for your comment; the information is described in Table 1 and Supplementary Table 3. The p-values ​​for the studies, as well as the type of analysis performed, are described in Table 1. However, one qualitative study and one study do not have a p-value. Even so, we consider it important to include this data.

Comment 8: Finally, the main claim (lines 19-23) does not follow from the review.

Response 8: Thank you for pointing that out. We have removed the information, as the magazine does not require it.

Comment 9: Inappropriate keywords that do not match the content. Why racism and necropolitics without physical activity?

Response 9: Thank you for your comment regarding the keywords; they have been changed to others more appropriate to the proposal.

Comment 10: A paragraph in Portuguese (lines 110-113).

Response 10: Thank you for pointing this out, and we apologize for the inconvenience. The session has been duly corrected in English.

Comment 11: Errors in the bibliography, despite the use of Zotero, for example, in [37] two first authors combined.

Response 11: Thank you for your feedback. We have made the necessary corrections.

Comment 13: Overall, the list of references does not meet healthcare standards.

Response 12: Thank you for your feedback. We have made the necessary corrections.

Comment 13: Occasional typographical errors, including in the appendix tables.

Response 13: Thank you for your comments. We have added the information to Supplementary Tables 1, 2, 3, and 4.

Round 2

Reviewer 1 Report

Comments and Suggestions for Authors

No further comments.

Author Response

Reviewer 1

Comments and suggestions for Authors

No further comments.

Response 1: We thank the reviewer for their comment.

Reviewer 3 Report

Comments and Suggestions for Authors

The new version of the article is much better than the previous one. 

I have two major remarks.

Firstly, the authors use three terms interchangeably in relation to health inequalities: inequities, inequalities, and disparities. Each of these terms has a different connotation and is understood slightly differently in different cultures. The term inequities, which appears in the title, emphasises social injustice. However, the keywords include disparities, which are considered the mildest term. I therefore suggest adding a paragraph to the introduction defining inequality and the terminology used in this review. It would be best to avoid using different terms interchangeably.

Secondly, the supplementary material now only contains a Word file with the same text as the PDF. The previous version included two extensive tables characterising the selected papers (samples defined as dyads and PA measurement methods). As I understand it, the current version has slightly reduced the overall number of papers described, so a new annex is needed. Furthermore, there is now a reference to four not two tables, but I do not have access to them. The tables are cited without maintaining a consistent order with the numbering. For example, table 4 from the appendix is cited first. This needs to be sorted out.

The description of selected studies (&3.2) lacks some important data from the supplement, for example on sample sizes.

Other minor comments:

Please, include in the abstract the initial number of papers found (4.195 studies), as this demonstrates the enormous amount of work that has been done.

In the footnote below Table 1, there is RN, and inside the table, there is NR (not reported).

Please, check the abstract. Finally, 26 or 25 studies?

Also the result:  Among these, 18 studies reported that higher parental/guardian PA was associated with greater child PA should be verified. 

It is worth adding an explanation below the table as to what significance the last column refers to. I have a more serious comment here, because in about 5/25 cases, there is ‘no’. In turn, the abstract contains a sentence (This review demonstrates consistent associations between parental/guardian and child PA) that is not included in the conclusions in the main text. Does this mean that the conclusion from the abstract contradicts the data in the table?

Author Response

Reviewer 3

Comments and Suggestions for Authors

The new version of the article is much better than the previous one. 

We appreciate the recognition of the manuscript's progress, and this would not have been possible without the valuable insights that were provided.

I have two major remarks.

Comments 1: Firstly, the authors use three terms interchangeably in relation to health inequalities: inequities, inequalities, and disparities. Each of these terms has a different connotation and is understood slightly differently in different cultures. The term inequities, which appears in the title, emphasises social injustice. However, the keywords include disparities, which are considered the mildest term. I therefore suggest adding a paragraph to the introduction defining inequality and the terminology used in this review. It would be best to avoid using different terms interchangeably.

Response 1: We thank the reviewer for this insightful comment. To avoid conceptual overlap and improve clarity, we removed all occurrences of the term “disparities” throughout the manuscript. In the Methods section, we now provide a concise explanation distinguishing difference from inequity, as this section offers a more technical and appropriate space to clarify these analytical distinctions without disrupting the narrative flow of the Introduction.

Although disparity, inequality, and inequity are often used interchangeably in the literature, we recognize that they may represent related but distinct concepts. However, discussing all three in detail is not central to the aim of this review. For this reason, we kept the Introduction focused on the construct of inequity, which aligns directly with our research objective, while refining the methodological description to prevent conceptual ambiguity.

Following the reviewer’s suggestion, we also removed “disparities” from the list of keywords and replaced it with “physical activity.” (Line: 135-138).

Comments 2: Secondly, the supplementary material now only contains a Word file with the same text as the PDF. The previous version included two extensive tables characterising the selected papers (samples defined as dyads and PA measurement methods). As I understand it, the current version has slightly reduced the overall number of papers described, so a new annex is needed. Furthermore, there is now a reference to four not two tables, but I do not have access to them. The tables are cited without maintaining a consistent order with the numbering. For example, table 4 from the appendix is cited first. This needs to be sorted out.

The description of selected studies (&3.2) lacks some important data from the supplement, for example on sample sizes.

Response 2: We sincerely apologize for the confusion caused by the previous upload of the supplementary material. It appears that an incorrect version of the file was submitted, which resulted in missing tables and inconsistent numbering. We appreciate the reviewer for drawing our attention to this issue.

The revised supplementary document now includes the four tables referenced in the manuscript, presented in the correct numerical order and aligned with the sequence in which they are cited in the text:

Supplementary Table 1: Descriptive summary of the PAGER framework applied to this scoping review on parental/guardian–child PA in relation to racial/ethnic inequities in the Americas.

Supplementary Table 2: Descriptive information for all included studies (objectives, methods, and main findings). This table also includes additional details on the study results, methodological specifications, and analytical focus relevant to the review.

Supplementary Table 3: Characteristics of study participants (authors and year, number of dyads, parent/guardian sample, child sample, average ages, caregiver type, and sex/age group of children/adolescents). We also added descriptive information on the sample size, along with summarized dyads; the tables contain more detailed information. (P6; Line: 263-265).

Supplementary Table 4: Detailed information on measurement methods for parental and child PA (domain, weekly frequency, duration, intensity, measurement instruments, data collection procedures, number of days, and specific questions/cutoff points).

These supplementary materials substantially expand the contextual information available, enabling a more comprehensive mapping of the methodological diversity and findings across studies—consistent with the purpose of a scoping review. Although such detail cannot be fully incorporated into the main manuscript, the supplementary document now provides a robust and coherent extension of the primary text.

Other minor comments:

Comments 3: Please, include in the abstract the initial number of papers found (4.195 studies), as this demonstrates the enormous amount of work that has been done.

Response 3: We thank the reviewer for this helpful suggestion. The initial number of records identified in the search (4,195 studies) has now been added to the abstract to better convey the scope of the screening process and the extensive work undertaken (P1; Line: 18).

Comments 4: In the footnote below Table 1, there is RN, and inside the table, there is NR (not reported).

Response 4: We thank you once again for your attention to the matter; the information has already been corrected.

Comments 5: Please, check the abstract. Finally, 26 or 25 studies?

Response 5: Thank you for raising this point. After rechecking all included articles during the latest revision to ensure full data consistency, we identified that one study (Ross et al., 2018) did not meet the inclusion criteria. Although the study examined parental and child PA and reported race/ethnicity, it did not assess parent–child PA associations; instead, it analyzed associations among parents and among children across different generations.
For this reason, the study was removed from the review, and the final number of included studies is 25.
For transparency, the DOI of the excluded study is provided:
https://doi.org/10.1186/s12889-018-5266-3. 

Comments 6: Also the result:  Among these, 18 studies reported that higher parental/guardian PA was associated with greater child PA should be verified.

explanation below the table as to what significance the last column refers to. I have a more serious comment here, because in about 5/25 cases, there is ‘no’. In turn, the abstract contains a sentence (This review demonstrates consistent associations between parental/guardian and child PA) that is not included in the conclusions in the main text. Does this mean that the conclusion from the abstract contradicts the data in the table?

Response 6: We thank the reviewer for this important observation. A minor inconsistency in the previous table description may have caused confusion, and we have now revised the table to ensure full clarity. Specifically, the binary classification (“Yes/No”) was insufficient to capture all types of findings, particularly:

Wen (2015) reported mixed results, with one domain showing a positive association and another showing no association, which we now classify as Null (inconsistent findings).

Nichols (2018) reported an inverse association for vigorous parental PA, which is now clearly labeled as Inverse.

To avoid ambiguity, we have expanded the table notes to read:

“Significant association” (Yes) refers to studies reporting at least one statistically significant association between parental/guardian PA and child PA (p < 0.05). Studies in which all reported associations had p ≥ 0.05 were classified as “No”. One study presented inconsistent findings (significant for one PA domain but non-significant for others) and was conservatively classified as “Null”. “Inverse” indicates a significant negative association.

In addition, we revised the abstract and the conclusion to ensure full alignment with the results. The statement that the review demonstrates “consistent associations” remains accurate, as the majority of studies (18 out of 25) reported positive associations; however, the presence of null and inverse results is now explicitly acknowledged in both the Results and Discussion sections, ensuring transparency and consistency.